# Factors Preventing Students with Disabilities from Participating in Sports at Rural Universities in Limpopo Province

**DOI:** 10.3390/ijerph22091370

**Published:** 2025-08-31

**Authors:** Tobias Johannes Mokwena, Takalani Grace Tshitangano, Shonisani Elizabeth Tshivhase

**Affiliations:** 1Department of Public Health, University of Venda, Thohoyandou 0950, South Africa; shonisani.tshivhase@univen.ac.za; 2Department of Public Health and Health Systems, University of Limpopo, Mankweng 0727, South Africa; takalani.tshitangano@ul.ac.za

**Keywords:** challenges, people with disabilities, societal barriers, sports participation

## Abstract

Students with disabilities face challenges that limit their participation in sports activities designed to promote social cohesion. This study examined factors that discourage students with disabilities from engaging in sports at rural universities in Limpopo Province. A qualitative research approach with an exploratory design was employed. Purposive sampling was used to select participants. Trustworthiness was ensured through measures of credibility, confirmability, transferability, and dependability. Credibility was established through prolonged engagement with participants during the data collection process. Confirmability was maintained by transcribing interview recordings verbatim without alterations. Transferability was supported by employing an appropriate study approach, design, target population, and inclusion criteria. Dependability was ensured by minimizing bias in participant selection. Data were collected through semi-structured interviews with sixteen students with disabilities, ten staff members from disability units at rural universities, and three staff members from sports and recreation units. Ethical principles-including informed consent, anonymity, privacy, confidentiality, and harm prevention-were consistently upheld throughout the study. Data analysis was conducted using a thematic reflexive analysis approach. Five main themes emerged: perceptions of including students with disabilities in sports programs; barriers preventing their participation; university support for students with disabilities interested in sports; the importance of engaging in sports programs; and interventions to encourage participation among students with disabilities. People with disabilities experience social exclusion, discrimination, and a lack of trained staff to support them in sports. The development and implementation of inclusive sports facilities can help address the challenges that prevent people with disabilities from participating in sports programs. An intersectoral approach is needed to ensure that people with disabilities participate in sports activities that promote a healthy lifestyle at universities.

## 1. Introduction

People with disabilities face numerous challenges that prevent them from engaging in sports programs designed to promote a healthy lifestyle [1]. According to the World Health Organization [2], people should participate in physical activities that promote health. However, few people with disabilities participate in sports programs compared to non-disabled people [3]. This notion is derived from the isolation, bullying, and discrimination of people with disabilities when interacting with non-disabled people [4].

It has been stated that people have the right to participate in activities that support physical development [5]. However, despite the significant health benefits of sports programs in addressing hypertension, cancer, and overweight issues, students with disabilities encounter barriers that hinder their participation in sports [6]. These barriers increase their risk of obesity and overweight, conditions linked to health problems that can be reduced through sports participation and a balanced diet [7]. This is a global public health concern that needs the intervention of various stakeholders to ensure that people participate in sports programs and address non-communicable diseases.

A study conducted in Lesotho reported that visually impaired students do not participate in sports activities due to a lack of sports facilities and adaptive sports programs to meet their needs [8]. In South Africa, there are few sports facilities designed for people with disabilities [9]. This remains an ongoing problem that requires various stakeholders to develop inclusive sports programs to encourage students with disabilities to participate in sports. The provision of sports facilities and programs plays a significant role in encouraging people to engage in sports and address a sedentary lifestyle [10].

## 2. Research Question

❖What are the personal barriers discouraging people with disabilities from participating in sports activities at rural universities in Limpopo Province?❖What are the architectural barriers discouraging people with disabilities from participating in sports activities at rural universities in Limpopo Province?

## 3. Theoretical Framework

According to Plooy et al. (2014) [11], a theoretical framework is a specific set of theories and ideas about the topic being studied. The study adopted the Health Promotion Model (HPM) as its framework. According to the World Health Organization, the Health Promotion Model focuses on creating an environment that enables people to take action on issues affecting their health [1]. The Health Promotion Model comprises various health promotion constructs such as perceived benefits of action, perceived barriers to action, self-efficacy, activity-related affect, interpersonal influences, situational influences, and commitment to a plan of action [12]. All these constructs within the Health Promotion Model play a vital role in promoting health.

### 3.1. Perceived Benefits of Action

The Health Promotion model plays a significant role in empowering people to participate in sports programs designed to promote well-being. The study of factors preventing students with disabilities from engaging in sports raises awareness and encourages people with disabilities to get involved. Participating in sports can also help reduce non-communicable diseases that affect individuals and enhance a healthy lifestyle.

### 3.2. Perceived Barriers to Action

The exploration of factors preventing students with disabilities from participating in sports will empower relevant stakeholders to develop and implement policies that will encourage people with disabilities to engage in sports. These policies might focus on monitoring and evaluation to ensure that there are adaptive sports facilities and programs for people with disabilities in the institutions of higher learning.

### 3.3. Self-Efficacy

When people are provided with information about the significance of engaging in sports and the disadvantages of not engaging in sports, they are likely to decide to participate in sports.

### 3.4. Activity-Related Affect

Activity-related affect means that people can be motivated to participate in sports programs designed to enhance a healthy lifestyle when they receive support from the university and other stakeholders. The provision of support for people with disabilities in sports can encourage sports participation.

### 3.5. Interpersonal Influences

Interpersonal influences are the relationships between people with disabilities and non-disabled people [13]. The University of Vend and the University of Limpopo have the responsibility to create a conducive environment where non-disabled people can interact with people with disabilities. A situation of this nature will encourage people with disabilities to participate in sports because sports participation results in social cohesion.

### 3.6. Situational Influences

When the University of Limpopo and the University of Venda promote a healthy lifestyle through the provision of sports facilities and adaptive programs, these will encourage students with disabilities to participate in sports programs. However, when the universities provide sports facilities that are not conducive to people with disabilities, this will discourage them from participating in sports.

### 3.7. Commitment to the Plan of Action

Universities, the Department of Higher Education, and Special Olympics South Africa should collaborate to make sure that people with disabilities participate in sports.

## 4. Materials and Methods

### 4.1. Study Design

The study employed a qualitative research approach. The study used an exploratory design to understand factors preventing students with disabilities from participating in sports. The design assisted in contributing insightful information about challenges experienced by people with disabilities in society. This research approach and study design were applied to 16 students with disabilities (8 male and 8 female), 10 universities’ disability unit staff, and 3 sports and recreation unit staff to explore the factors discouraging students with disabilities from participating in sports activities at universities in Limpopo Province. The exploratory research approach enabled the researchers to gain insight into the challenges experienced by people with disabilities when attempting to engage in sports.

### 4.2. Study Setting

The study setting is the environment where the study was conducted [14]. The study was conducted at the University of Venda and the University of Limpopo, which are in Limpopo Province. Limpopo province is among the rural provinces in South Africa, characterized by several mining companies. The University of Limpopo and the University of Venda have the Department of Sport to provide and regulate sports activities. Both universities have disability centers to assist students with disabilities in academic programs and provide social support.

### 4.3. Inclusion and Exclusion Criteria

The study included undergraduate students with disabilities from the University of Venda and the University of Limpopo who were 18 years and above. Students who are intellectually impaired, physically disabled, visually impaired, and hearing impaired were included in the study. Rural-based university staff in Limpopo Province who were working directly with students with disabilities were included in the study. Those who did not consent were excluded from participating in the study.

### 4.4. Study Sampling

Purposive sampling was adopted to select participants who voluntarily participated in the study until data saturation was reached. The study reached data saturation when the participants no longer provided new information during data collection. The participants were repeating information that had been shared by previous participants.

### 4.5. Inter-Rater Reliability

The study pretested an interview guide on the students and the university staff before data collection. The data collected were analyzed, and the authors reached an agreement that the data collection instrument can be used to collect the data.

### 4.6. Data Collection

The study used a semi-structured interview guide for participants who were willing to participate. Participants were provided with information about the study before participating. Participants were advised that they were participating in the study voluntarily. Those who wished to withdraw from participating could do so at any time. Participants were between 18 and 59 years old. The interview guide was presented to the study supervisors, departmental seminars, and the University of Venda Higher Degree Committee before data collection to maintain trustworthiness. Trustworthiness was achieved through credibility, confirmability, transferability, and dependability. Credibility was ensured through having prolonged engagement with the participants during data collection. To ensure transferability, the study described the participants and the methodology followed. Confirmability was achieved through transcribing the interview records without alteration. Dependability was achieved by ensuring that no bias influenced people to participate in the study. Data was collected after the participants completed a written consent form. Participants who did not consent were excluded. An audio tape recorder, which lasted approximately 45 min per participant, was used to record the interview.

### 4.7. Data Analysis

Thematic reflexive analysis was used to analyze the data. The researcher played the audio recordings repeatedly and reviewed the transcripts to ensure that there was no information that was lost. The study organized the data into different categories by having a general comprehension of the information that had been gathered. Themes and sub-themes were developed based on the respondents’ remarks. The researcher clustered similarities into themes and sub-themes. The study findings were then evaluated, explaining the lessons learned, which captured the essence of the idea.

## 5. Results

Five main themes emerged from the analyzed data, namely, perceptions about the inclusion of students with disabilities in sports programs, barriers discouraging people with disabilities from engaging in sports programs, universities’ conducive sports facilities and programs for students with disabilities, universities’ ability to provide support for students with disabilities who want to engage in sports, the significance of engaging in sports programs, and interventions for encouraging students with disabilities to engage in sports programs. A total of 29 participants participated in the study (see Table 1 and Table 2).

Students who participated in the study ranged between 18 and 30 years. All sixteen (16) participants were black undergraduate students with disabilities from the University of Venda and the University of Limpopo. Students who were physically disabled, epileptic, visually impaired, hearing impaired, and intellectually disabled were interviewed in the presence of the disability unit staff (see Table 1).

The university staff participants ranged between 18 and 59 years. All participants, thirteen (13), were black. Most of the participants, eight, were not married. Most of the participants (twelve) were Christians; however, only one participant was affiliated with African Traditional Religion (ATR). Most of the participants (10) were attached to the University of Limpopo. Only three (03) were attached to the University of Venda (See Table 2).

### 5.1. Perceptions About the Inclusion of Students with Disabilities in Sports Programs

The study examined the inclusion of students with disabilities in sports programs at universities in Limpopo Province. The findings indicated that students have different views about the inclusion of students with disabilities in sports. The respondents indicated that students with disabilities do not receive the same treatment as non-disabled students in sports, while some emphasized that there are sports programs at universities. However, these programs do not accommodate students with disabilities. The statement below indicates how students with disabilities are excluded from sports programs:


*“All students were included in all different sporting codes, but as time goes on, we are no longer included in many sporting codes. Today, we find that there is only one sports activity for people with disabilities, which is goalball, is the one that is active now. So, it means we are not included in any sports.”*

*(Participant 18, Individual Interview)*


Another respondent shared a sentiment about the exclusion of students with disabilities. This is reflected in the statement:


*“Normally, I must use a scooter to go to class. So, when I’m walking around, non-disabled students discriminate against us and give us a negative attitude, which results in low self-esteem. They see us as different people, and the lecture halls are not accommodating people with disability.”*

*(Participant 2, Individual Interview)*


Another respondent shared a sentiment about the exclusion of students with disabilities in sports. This is reflected in the statement:


*“We went to different secondary schools, and we are coming from different environments, different areas, and different provinces. Most people faced bullying during secondary school because of their disabilities, yeah. We are being labelled that disabled person cannot do these. Most of the disabled students allowed those words and prevented them from participating in sports activities.”*

*(Participant 8, Individual Interview)*


### 5.2. Barriers Discouraging People with Disabilities from Engaging in Sports Programs

Evidence from the qualitative interviews indicated that students with disabilities faced challenges discouraging them from engaging in sports programs. The statement below indicates that students with disabilities experience challenges that prevent them from participating in sports:


*“No, because I have looked around those sports facilities and sports activities that general students participate in. I have seen that it accommodates mostly students without disability. I could see that there is no way that someone who is disabled can participate well in those facilities.”*

*(Participant 5, Individual Interview)*


Another respondent shared a similar sentiment about the challenges discouraging students with disabilities from participating in sports programs:


*“Another factor is that we share the sports hall, which is being shared with other sporting codes, so if we must do proper markings, which are needed or essential for goalball players to be able to navigate around the court. We find that the next day, the other sporting code people have removed them because it was disturbing them, or it disturbed them.”*

*(Participant 1, Individual Interview)*


Respondents indicated that non-disabled students have negative attitudes toward students with disabilities. The statement below indicates challenges experienced by people with disabilities:


*“I do not participate in sports activities at the campus because non-disabled students tend to isolate us, and such behavior affects us emotionally. I attempted to participate in sports activities; however, non-disabled students are not welcoming us. That is the reason I do not participate in sports activities.”*

*(Participant 16, Individual Interview)*


Another respondent indicated that the knowledge and skills about adaptive sports programs play a significant role in providing adaptive sports activities


*“I do not think they are discriminated. There is only that, as a university, we do not have professionals who are knowledgeable about the sporting activities for people with disabilities. What I know is that they teach themselves without the intervention of a professional or someone who has knowledge about disability sports activities. I think in the department itself, we do not have a person who’s knowledgeable about the different sporting codes that students with disability can participate in.”*

*(Participant 18, Individual Interview)*


### 5.3. The Universities Support Adaptive Sports Programs

A respondent asserted that there is a lack of support in adaptive sports programs from the university’s management. The lack of support in adaptive sports programs exposes students with disabilities to an unhealthy lifestyle. The statement below indicates that the university provides support to people with disabilities:


*“The support is there, but I don’t think it’s enough. I believe the university can do more to make it easier for students to access sports facilities, because it seems that for everything that students are interested in, the focus is more on administration than implementation.”*

*(Participant 16, Individual Interview)*


### 5.4. The Significance of Engaging in Sports Activities

Respondents indicated that there are people who know the significance of engaging in sports, and there are those who don’t know the significance of engaging in sports. A respondent suggested that sports participation improves blood circulation


*“Participating in sports activities helps to improve blood circulation. Sports make somebody fit and healthy.”*

*(Participant 19, Individual Interview)*


Another respondent indicated that sports participation plays a significant role in improving social health. This is an effective mechanism for addressing the antisocial behavior displayed by some people with disabilities and non-disabled people:


*“The benefits of participating in sports include exploring the world. You can interact with different people. It helps your body. It gives you good mental health. You can communicate with others and make friends. The more you network with others who are not disabled, you become free yourself.”*

*(Participant 1, Individual Interview)*


### 5.5. Interventions to Encourage Students with Disabilities to Engage in Sports Programs

Respondents indicated that the universities should recruit qualified professional coaches who are knowledgeable about adaptive sports programs for people with disabilities to participate in sports:


*“The first thing is that the department of sports and student affairs should hire people who know people with disabilities, that is the starting point. Then, the person will motivate and encourage people with disabilities to participate in sports activities.”*

*(Participant 18, Individual Interview)*


Respondents indicated that there is a need to establish new sports bodies that accommodate adaptive sports programs:


*“I think the university is implementing measures such as introducing new sports bodies next year to recruit first-year students to join the sports activities that are available at the campus.”*

*(Participant 22, Individual Interview)*


Respondents emphasized a need to implement new adaptive sports programs and revive old adaptive sports activities to encourage students with disabilities to participate in sports:


*“There was chess in the campus. I think if it can be brought back, students with disabilities can participate. There are also these other sports where they use wheelchairs, I don’t know the name of the sports, and I think students who are using wheelchairs can participate in that one.”*

*(Participant 17, Individual Interview)*


Respondents indicated that there is a need to organize a meeting and introduce sports programs to students:


*“I think they can call a meeting, where they will be introducing all of us to different sports activities available on the campus. This will help us choose sports activities based on the nature of the disability. The university should organize a mini function where they can invite all students with disabilities and then start teaching them about sports, or maybe have an online platform where they will be announcing different sports activities and ensuring that people with disabilities are accommodated.”*

*(Participant 8, Individual Interview)*


The findings indicated that it is important to organize annual sports events that comprise adaptive sports programs


*“I think they must start an awareness campaign encouraging students living with disabilities to participate in sports.”*

*(Participant 22, Individual Interview)*


Respondents indicated that the universities should liaise with special schools where most of the students with disabilities studied to provide effective adaptive sports programs. alignment with special schools will assist the universities in developing adaptive sports programs based on the disabilities of the people:


*“As the unit of sports and recreation, we must liaise with special schools where most of our students with disabilities come, so that they can be exposed to most sports activities. Sports should be prioritized at special schools so that our work can be simple.”*

*(Participant 29, Individual Interview)*


Respondents suggested that universities should provide adaptive sports equipment to encourage students with disabilities to participate in sports programs:


*“I think the university should provide proper equipment for sports that are already offered at the campus and try to introduce one or more sports activities that can cater to people with disability.”*

*(Participant 15, Individual Interview)*


## 6. Discussion

This study examined factors that prevent students with disabilities from participating in sports at rural universities in Limpopo Province.

### 6.1. Perceptions About the Inclusion of Students with Disabilities in Sports Programs

The findings indicated that students with disabilities are excluded from sports programs designed for social cohesion. The findings concur with a study conducted by Breffka et al. [1], who asserted that people with disabilities experience multiple challenges that prevent them from participating in sports programs designed for social cohesion. Similar findings indicated that people with disabilities experience isolation, bullying, and discrimination when interacting with non-disabled people [4]. Sports play a significant role in addressing hypertension, cancer, and being overweight [6]. The challenges experienced by people with disabilities prevent them from engaging in activities designed to address non-communicable diseases and health conditions such as hypertension, cancer, obesity, and overweight. This is a societal problem that needs various stakeholders to ensure people with disabilities participate in sports programs designed to enhance a healthy lifestyle.

### 6.2. Barriers Discouraging People with Disabilities from Engaging in Sports Activities

The findings indicated that students with disabilities experience challenges in accessing sports infrastructure when attempting to participate in sports programs. The findings are similar to a study conducted by Carbone et al. [6], who reported that despite the significant health benefits of participating in sports programs to address hypertension, cancer, and overweight, people with disabilities experience challenges that prevent them from participating in sports programs. The findings aligned with a study conducted by Makwela et al. [9], who alluded that people with disabilities experience challenges in accessing sports infrastructure, bullying, and a lack of adaptive equipment when participating in sports. The lack of adaptive sports equipment and infrastructure that is user-friendly to people with disabilities excludes them from engaging in sports programs designed for social cohesion and personal development. This is a societal challenge that requires urgent measures to ensure that people with disabilities enjoy the significance of participating in sports.

### 6.3. Support to Participate in Sports Programs

The findings indicated that students with disabilities do not receive support to participate in sports programs. The findings are similar to a study conducted in Lesotho by Ralejoe [8], who asserted that people who are visually impaired do not participate in sports activities because of a lack of sports facilities and adaptive sports programs. This is common in rural areas because of a lack of resources and financial constraints. A situation of this nature compromises the well-being of people with disabilities. This is an urgent concern that needs the collaboration of various experts to ensure that people with disabilities participate in sports. Contrary findings revealed that people with disabilities should be provided with support to participate in sports programs [5]. It is the responsibility of the Department of Sports, Arts and Culture, municipalities, and sports federations to provide support for people with disabilities to participate in sports. This will help address the misinformation about visually impaired people and the discrimination and bullying that people with disabilities experience in different aspects of life.

### 6.4. The Significance of Engaging in Sports Activities and the Interventions to Encourage Students with Disabilities to Engage in Sports Activities

The findings indicated that sports participation helps improve blood circulation, physical health, mental health, and social health. The study concurs with a study conducted by Carbone et al. [6], who asserted that sports participation helps address hypertension and overweight. This will help address the non-communicable diseases in society, which is in line with Sustainable Development Goal 3. This shows that sports play a significant role in people’s well-being.

### 6.5. Interventions to Encourage People with Disabilities to Participate in Sports

The findings showed that various stakeholders need to develop inclusive policies that encourage people with disabilities to participate in sports. The study concurred with a study conducted by Deng et al. [10], who indicated that the provision of sports facilities and programs plays a significant role in encouraging people to engage in sports and address a sedentary lifestyle. Creating and implementing inclusive sports facilities can help address the challenges that prevent people with disabilities from engaging in sports programs.

## 7. Limitations of the Study

The study focused on the University of Venda and the University of Limpopo, which are rural-based universities. Therefore, the study cannot be generalized to urban-based universities. The findings may not apply to other provinces in South Africa because of the different sports infrastructure.

## 8. Conclusions

People with disabilities experience multiple challenges preventing them from participating in sports programs designed to enhance a healthy lifestyle. These expose people with disabilities to non-communicable diseases such as hypertension, overweight, and obesity. To address this concern, various stakeholders should collaborate to develop inclusive sports programs to encourage students with disabilities to engage in sports.

## Figures and Tables

**Table 1 ijerph-22-01370-t001:** Demographic information on students with disabilities.

Codes	Age	Institution	Gender	Race	Type of Disability	Educational Level	Sports Participation
Participant 1	25	University of Venda	Male	Black	Physical disability	Third Year	Yes
Participant 2	22	University of Venda	Female	Black	Physical disability	First Year	No
Participant 3	19	University of Venda	Female	Black	Physical disability	First Year	Yes
Participant 4	25	University of Venda	Female	Black	Epilepsy	Fourth Year	Yes
Participant 5	19	University of Venda	Female	Black	Visually Impaired	Second Year	No
Participant 6	23	University of Venda	Male	Black	Hearing Impaired	First Year	No
Participant 7	18	University of Venda	Female	Black	Visually Impaired	First Year	No
Participant 8	24	University of Venda	Female	Black	Hearing Impaired	Second Year	No
Participant 9	23	University of Venda	Male	Black	Other	First Year	No
Participant 10	26	University of Venda	Male	Black	Physical disability	Second Year	No
Participant 11	21	University of Venda	Male	Black	Physical disability	Second Year	Yes
Participant 12	29	University of Limpopo	Female	Black	Physical disability	Fourth Year	Yes
Participant 13	25	University of Limpopo	Male	Black	Physical disability	Third Year	No
Participant 14	21	University of Limpopo	Female	Black	Physical disability	Fourth Year	No
Participant 15	23	University of Limpopo	Male	Black	Physical disability	First Year	Yes
Participant 16	30	University of Limpopo	Male	Black	Physical disability	Fourth Year	Yes

**Table 2 ijerph-22-01370-t002:** Demographic information for rural-based university staff.

Codes	Age Group	Institution	Gender	Race	Marital Status	Educational Level	Religion
Participant 17	50–59	University of Limpopo	Female	Black	Married	PhD	Christianity
Participant 18	50–59	University of Venda	Male	Black	Married	Honors	Christianity
Participant 19	29–39	University of Venda	Female	Black	Single	Masters	Christianity
Participant 20	18–28	University Limpopo	Male	Black	Single	Masters	African Religion
Participant 21	29–39	University of Limpopo	Female	Black	Single	Degree	Christianity
Participant 22	29–39	University of Limpopo	Male	Black	Single	Degree	Christianity
Participant 23	18–28	University of Limpopo	Female	Black	Single	Honors	Christianity
Participant 24	40–49	University of Limpopo	Female	Black	Married	PhD	Christianity
Participant 25	18–28	University of Limpopo	Male	Black	Single	Masters	Christianity
Participant 26	29–39	University of Limpopo	Male	Black	Single	Masters	Christianity
Participant 27	50–59	University of Limpopo	Female	Black	Separated	Masters	Christianity
Participant 28	50–59	University of Limpopo	Female	Black	Single	Masters	Christianity
Participant 29	29–39	University of Venda	Female	Black	Married	Honors	Christianity

## Data Availability

The data is available on request from tobiasmokoena@gmail.com.

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
