# Peer review of "Factors Preventing Students with Disabilities from Participating in Sports at Rural Universities in Limpopo Province"

_ijerph, 2025, doi:10.3390/ijerph22091370_

Round 1
Reviewer 1 Report
Comments and Suggestions for Authors
Peer Review Report
Title: Factors Preventing Students with Disabilities from Participating in Sports at Rural Universities in Limpopo Province
General Comments
This qualitative study investigates the obstacles students with disabilities encounter when trying to participate in sports at rural universities in Limpopo Province. While the subject matter is undeniably important and not extensively researched, the current manuscript has notable weaknesses in its methodology, analysis, and presentation, which significantly diminish its potential contribution to the existing literature.
Major Weaknesses
Methodological Rigor: The study provides insufficient detail regarding crucial aspects such as sampling procedures, how data saturation was determined, and the measures taken to ensure trustworthiness. The absence of a discussion on reflexivity, a lack of information on member checking procedures, and the omission of triangulation strategies all raise serious questions about the study's credibility and dependability.
Data Analysis Limitations: The thematic analysis process is not adequately described. There's no information on the specific coding procedures used, how inter-rater reliability was addressed, or the methods employed for theme validation. The analysis itself appears rather superficial, lacking depth in the development of themes and failing to integrate theoretical concepts effectively.
Literature Foundation: The literature review is notably constrained. It overlooks key references within the field of disability sport and does not succeed in establishing a comprehensive theoretical basis for the study.
Minor Weaknesses
Writing Quality: Throughout the manuscript, persistent grammatical errors, inconsistent terminology, and poorly structured sentences necessitate considerable revision.
Structural Organization: Several sections, particularly the results and discussion, lack a logical flow and clear organization.
Specific Comments
Abstract
Line 13-14, Page 1: The statement "The study employed purposive sampling" needs further clarification regarding the specific sampling strategy and the criteria used.
Line 19-20, Page 1: "Five main themes emerged" – this statement should specify the analytical framework that was utilized to identify these themes.
Introduction
Line 30-32, Page 1: The opening statement "Sport plays a crucial role in addressing the challenges" is too vague and requires more precise articulation.
Line 33-35, Page 1: The UNESCO citation appears to be incomplete and needs to be properly formatted according to the journal's guidelines.
Line 36-40, Page 1: This sentence is excessively long and contains grammatical errors that make it difficult to comprehend.
Research Questions
Line 52-58, Page 2: The three research questions are somewhat repetitive and could benefit from consolidation into a more cohesive framework that addresses the barriers comprehensively.
Theoretical Framework
Line 60-67, Page 2: The explanation of the Health Promotion Model is superficial and does not adequately integrate with the study's specific context and objectives.
Methods
Line 68-78, Page 2: The sampling description lacks essential details concerning recruitment procedures, inclusion/exclusion criteria, and how data saturation was determined.
Line 79-90, Page 2: The data collection section needs a more detailed description of how the interview guide was developed and the procedures for pilot testing.
Data Analysis
Line 91-97, Page 3: The description of the thematic analysis is inadequate, missing crucial information about the coding processes, any software used, and measures taken for inter-rater reliability.
Results
Line 106, Page 3: Table 1 contains formatting errors and inconsistent abbreviations (Univen, UL) that require standardization for clarity.
Line 113, Page 4: Table 2 exhibits similar formatting issues and unclear categorizations.
Line 128-132, Page 4: The participant quote needs a proper contextual introduction and analysis, rather than being presented in isolation.
Line 149-153, Page 5: This particular quote is excessively long and could be more effectively summarized while retaining its key insights.
Discussion
Line 255-264, Page 7: The discussion section lacks sufficient integration with existing literature and the theoretical framework.
Line 265-272, Page 7: The comparison with Ralejoe (2021) requires a more critical analysis and a deeper theoretical connection.
Limitations
Line 301-305, Page 8: The limitations section is too brief and does not adequately address the methodological limitations of the study.
References
Line 326-349, Page 9: The reference list shows formatting inconsistencies and appears incomplete, especially given the limited integration of literature throughout the manuscript.
Comments on the Quality of English LanguageThe manuscript is marred by numerous grammatical errors.
Reviewer 2 Report
Comments and Suggestions for Authors
This study used qualitative methods to identify factors that hinder the participation of students with disabilities in sports programs. The logic is consistent from the Introduction to the Conclusion, and ethical details (informed consent, anonymization, etc.) are clearly described. As a reviewer, I find the academic reliability to be high. To further improve this paper and make it applicable to promoting the participation of students with disabilities in sports in rural areas around the world, I propose the following two points:
- Strengthen the international comparative perspective
Currently, this paper focuses on a case study in Limpopo Province, making it difficult for readers to gauge the “universality” or “differences” of the identified barriers—such as “barriers preventing people with disabilities from participating in sports programs” or the level of “university support for students with disabilities interested in sports.” Incorporating comparative discussions with similar studies (e.g., international para-sports initiatives, WHO health‐promotion frameworks, UN case studies) will highlight where your findings converge with or diverge from global trends.
- Link findings to international policy frameworks
By integrating key provisions of the UN Convention on the Rights of Persons with Disabilities (CRPD)—notably Article 30 on participation in sports and recreation—and guidelines from the UN Sports for Development Program into your “interventions to encourage participation among students with disabilities” theme, you will situate your results within a broader policy context, thereby attracting a wider international audience.
Reviewer 3 Report
Comments and Suggestions for Authors
Your study explored factors preventing students with disabilities from engaging in sports at rural universities in Limpopo Province. This is an area that will add to literature as there is paucity in research around sports for students with disabilities in rural universities in South Africa and the world at large. It will be a good article for publication after you have done a major revision. The following are my recommendations:
- You have not provided any relevant literature about your study.
- Line 60, your study adopted Health Promotion Model (HPM) as a framework. You explained what HPM was, but you did not indicate how this theory supported your study
- Your study identified five themes namely, perceptions of including students with disabilities in sports programs, barriers preventing people with disabilities from participating in sports, university support for students with disabilities interested in sports, the importance of engaging in sports activities and interventions to encourage participation among students with disabilities. Under your discussion segment, you discussed themes 1, 2 and 4. Others were not discussed.
- You combined themes 4 and 5 but couldn’t see the discussion around theme 5 (i.e., the interventions).
- Kindly discuss the university support for students with disabilities interested in sports
- Also take note of the following:
- Line 47: The study findings further indicated that [should read “The findings further … OR the study further …”
- The term “students living with disabilities” e.g., see lines 107, 110, 175, 205, 219 etc should be avoided. Just write “students with disabilities”
- Line 236, 255, 256, 265 … as indicated earlier, don’t say the study findings, but say “the findings …” OR “the study ….”
- Line 299 should be written as Sustainable Development Goal 3
- The acknowledgement needs to be rewritten: For example, you wrote: “We want to thank the University of Venda for issuing ethical clearance” I don’t think you can thank the university. You may thank your VC, HoD, the chair and members of the ethical committee, lecturers etc in the University of Venda. You also talked about Mr. Duppy Manyuma. What has he done about the manuscript? Reading through the manuscript, giving advice, encouraging you etc must be indicated
- Once you add literature review, your references will increase from 9 as it is now. You talked about UNESCO report, but this was not seen in your reference list
Round 2
Reviewer 1 Report
Comments and Suggestions for Authors
GENERAL COMMENTS
The authors have done a good job addressing many of the previous concerns, and the research question itself is both timely and important. The focus on barriers to sports participation for students with disabilities in rural university settings fills a notable gap. The qualitative approach is certainly a good fit for exploring a topic with such complex social dimensions. I've identified a few areas where the manuscript could be strengthened to reach the journal's standards. My main concerns are related to the methodological detail and the depth of the analysis. The description of the analytical procedures is currently a bit thin, and I'd like to see more details on how inter-rater reliability was handled and how the point of data saturation was determined. Additionally, I think the thematic development could be richer; I'm looking for a stronger integration of the participant quotes with the authors' analytical interpretation.
There are also a few minor points that, while not critical, should be addressed:
- Reference Formatting: The citation style seems inconsistent in a few places, and a couple of the references appear to be incomplete.
- Table Presentation: I noticed some formatting inconsistencies in the demographic tables that should be standardized.
- Discussion Depth: The discussion could benefit from a more critical analysis, and the limitations section feels a bit superficial.
SPECIFIC COMMENTS BY SECTION
Abstract:
Lines 16-17: The statement about trustworthiness measures is a bit too general. To make this part of the abstract more impactful, I would suggest specifying exactly how each criterion (credibility, confirmability, etc.) was actively maintained.
Line 21: The paper mentions that thematic analysis was used. It would be helpful to specify which particular approach to thematic analysis was employed (e.g., Braun & Clarke's, reflexive thematic analysis, etc.) to give readers a clearer picture of the methodology.
Introduction:
Line 36: The opening line is quite broad. I'd suggest starting with a more focused statement that immediately connects to the specific context of the study.
Lines 68-69: The sentence "is derived from isolation, bullying, and discrimination" has a small grammatical error. It should be rephrased to "derives from" or the sentence restructured entirely for better flow.
Lines 88-89: The formatting of the research question is inconsistent with standard academic conventions. It should be presented in a way that is clear and aligns with the journal's style guide.
Theoretical Framework:
Lines 103-104: The claim that people are more likely to participate in sports when they "recognize the importance" lacks a bit of theoretical and empirical backing. It would be good to ground this section in a more established theoretical framework.
Lines 266-268: The definition of "Activity-related affect" as "people have the responsibility to take care of their health" seems imprecise and doesn't quite align with the established constructs of the Health Promotion Model. This definition needs to be revised for accuracy.
Methodology:
Line 292: The manuscript mentions an "exploratory research design." I would recommend providing a bit more detail about the specific exploratory approach used.
Lines 344-345: The authors state that trustworthiness was ensured, which is great. Could they please be more specific about the concrete strategies used for each criterion (credibility, confirmability, etc.)?
Line 352: Again, describing the specific analytical framework and coding procedures used for the thematic analysis would enhance the transparency and rigor of the paper.
Results:
Table 1: The formatting in the demographic tables seems a little inconsistent, particularly with the italicization of institution names. It would be best to make this consistent.
Lines 738-741: The quote attribution "(p18, Individual Interview)" is a little ambiguous. Could the authors clarify if this refers to participant 18 or page 18 of the transcript?
Lines 879-887: There's a long quote here that needs to be better integrated with the surrounding text. I'd suggest adding more of the authors' own analysis to set up and follow up on the quote's significance.
Discussion:
Lines 999-1000: The phrase "The findings concur with..." is used multiple times throughout the discussion. I suggest varying the language to make this section more dynamic and engaging.
Line 1019: There's a simple spacing error here: "needsthe collaboration" should be "needs the collaboration."
Lines 1097-1100: The study limitations section could be improved with a more critical and reflective discussion of the methodological constraints and how they might have influenced the findings.
References:
Reference 4: The citation for Creswell, J. (2013) is incomplete. It's missing the full title and publisher details.
Reference 10: The formatting of this reference is inconsistent with the others in the list.
Comments on the Quality of English LanguageThere are a persistent grammatical errors, awkward phrasing, and inconsistent formatting throughout the manuscript compromise readability of the paper
Reviewer 3 Report
Comments and Suggestions for Authors
You addressed all recommendations I raised in the first review. Congratulations!
Author Response
Thank you.